# Design and Analysis of a 6-DOF Magnetic Suspension Platform with an Improved Permanent Magnetic Array

**DOI:** 10.3390/s22114067

**Published:** 2022-05-27

**Authors:** Shinan Cao, Pingjuan Niu, Jie Bai, Wei Wang, Qiang Liu, Sha Sheng, Jing Li

**Affiliations:** 1School of Mechanical Engineering, TianGong University, TianJin 300387, China; zsqjss@163.com (S.C.); juliabai2010@163.com (J.B.); 2Institute of Precision Electromagnetic Equipment and Advanced Measurement Technology, Beijing Institute of Petrochemical Technology, Beijing 102617, China; shaowang66@163.com (W.W.); liuqiangbuaa@163.com (Q.L.); shengsha@bipt.edu.cn (S.S.); bipt_lijing@bipt.edu.cn (J.L.)

**Keywords:** magnetic suspension platform, permanent magnetic array, driving force model, the Lorentz force characteristics, magnetic flux density, current stiffness

## Abstract

For ultra-precision, large stroke, and high start/stop acceleration, a novel 6-DOF magnetic suspension platform with a novel structure of the permanent array is proposed. The structure and the working principle of the novel platform are introduced. An accurate model of the novel structure was established to calculate the magnetic density distribution for obtaining the parameters and performance of the magnetic suspension platform. The analytical model’s results were verified by the finite element method. The driving force model of the magnetic suspension platform was established based on the Lorentz force. Twelve laser displacement sensors were applied to perceive the posture and vibration acceleration of the platform. The hardware information and the measurement models were introduced and established based on the layout. Finally, the Lorentz force characteristics of the proposed platform were investigated and compared with the conventional magnetic platform by the finite element analysis. The results show that the average magnetic flux density is 0.54T, the horizontal current stiffness along the *X*-axis is 63.1N/A, the current stiffness along the *Y*-axis is 61.6N/A, and the average output torque is 7.2 N*cm of the novel platform, larger than those of the conventional ones.

## 1. Introduction

High-precision platforms are widely used in the multidisciplinary nanotechnology field, the lithographic processing of semiconductors, precision machining, precision measurement, and mass transfer. The four types of platforms: mechanical platform [1], air-floating platform [2], liquid-floating platform [3], and magnetic suspension platform [4] can be used in high-precision. The mechanical platform is simple and easy to manufacture, but it has friction and vibration at high speeds, which limits its use in ultra-clean rooms. The air-floating platform can be controlled to move at a high velocity, but it has low precision and insufficient load capacity. The liquid-floating platform has high positioning accuracy and a high load-carrying capacity, but the velocity is low and it has a complex structure. As magnetic suspension technology is used in a magnetic suspension platform, the magnetic suspension platform has the following advantages: a compact structure, high transmission stiffness, a fast dynamic response, high positioning accuracy, and noncontact characteristics. It has received more attention in recent years.

In reference [5], the authors invented the first magnetic suspension platform; it has a linear motor that can provide force in both suspension and translation. However, its stroke is smaller, and the acceleration of the platform is only 1 g. Subsequently, many scholars began to design and analyze magnetic suspension platforms with different strokes and functions. In reference [6], the authors designed a triangular magnetic suspension platform, compared with the first magnetic suspension platform, the maximum acceleration is 2 g and the maximum velocity is 0.4 m/s. The calorific value is reduced in this platform. A Y-shaped platen was designed in reference [7]. The stroke of the Y-shape platen is ∓5 mm and the position resolution is 3 nm. Three permanent magnetics were used in this platform; thus, it has a simple structure. Moreover, the platform can recover stability in a short time when there is an external load. In reference [8], a 2-DOF electromagnetic actuator with a permanent and two windings was proposed. The rectangular and cylindrical windings are located below the axially-magnetized cylindrical permanent and are used to generate the horizontal and vertical driving forces. Reference [9] proposed an electromagnetic suspension platform. A large suspension force can be produced with a low current. The maximum velocity is 200 mm/s; however, the position accuracy is only 3 μm. So it cannot be used in an ultra-precision platform. In reference [10], different from the platforms above, the authors invented a magnetic suspension platform with a two-axis linear actuator. The designed travel volume of the stage is 2 × 2 × 2 mm in translation and 4° × 4° × 4° in rotation. The platform is very compact; three compact two-axis linear actuators and six power amplifiers were used in the platform. The precision is 1.1 nm for x, 0.74 nm for y, and 4.4 nm for z. In reference [11], to improve the position accuracy and the speed of the multi-axis stage, a surface motor-driven planar motion stage integrated with an XY $\theta_{z}$ surface encoder was proposed. The precision positioning can be carried out independently in X, Y, and $\theta_{z}$ with resolutions of 200 nm and 1″ by using the angle sensor. The stroke is 40 mm in the X and Y directions. The platform in reference [11] improves the positioning accuracy but only achieves 3-DOF. A 5-DOF planar motion stage is proposed in reference [12] by integrating the surface encoder. Three linear motors were used to generate movement of the stage in the XY $\theta_{z}$ directions and another two were used to move the stage in the $\theta_{x}\theta_{y}$ directions. The resolution is 30 nm, which is higher than in reference [11]; the resolution is 0.1″ in the $\theta_{x}$ and $\theta_{y}$ directions, which is also better than in reference [11]. A similar platform was designed in reference [13]. In reference [14], a platform with a single axis Lorentz force actuator was invented, which was driven by the pure Lorentz force. The travel volume is 2 × 2 × 2 mm in translation and 80 × 80 × 40 mrad in rotation. The resolution of the translation is 2.8 μm. Reference [15] proposed a platform with a 2-DOF electromagnetic actuator of an improved Halbach array. In reference [16], the vertical and horizontal driving forces were produced by two sets of circuit board conductors perpendicular to each other in the same permanent magnetic field. As the effective lengths of the conducting wires are short, so the platform’s stroke is short and the force is low.

All the platforms above have short strokes. To improve the platform’s stroke, in reference [17], a small-scale laboratory system was used to improve stroke and develop the technology required for the magnetic suspension of objects over large ranges of orientation. The rotation about the cylinder axis is not controlled, so the platform can be made to undergo a full 360°. In reference [18], a novel magnetic bearing using Halbach magnet arrays was used in the platform. By using the magnetic bearing, the mass of the platform is compensated, and the stroke can be increased four times in a vertical than reference [17]; however, the platform has a complex structure and insufficient stiffness. To solve this problem, references [19,20] designed a TU-shape platform. A suspension coil and propulsion coil were used. The minimum stroke was 10 × 10 × 10 mm; the stroke can be increased by increasing the length of the stator yoke. However, the utilization ratio of the current winding is only 30% and the power consumption is high. Reference [21] designed magnetically levitated planar actuators with moving Halbach permanent magnets. The coils are fixed in the stator and the stroke can be improved by increasing the lengths of the coils. As the magnetic moves when the platform works, it is hard for the stability of the control system with the actuators. In reference [22], A platform with three coil pairs was designed. There, Halbach arrays were used in the moving part (two or four Halbach arrays). Its stroke is 100 × 100 mm^2^ in the *x*, *y* directions and 100 μm in the *z*-direction. A same magnetic suspension platform is designed in reference [23]. A magnetic levitation stage system that has a unique motor structure fusing a gravity compensation function and pitching moment compensation is proposed in reference [24]. It has a coarse stage and a fine stage. The coarse stroke is ∓200 mm and the fine stroke is ∓3 mm. As two stages are used in the platform, it has a complex structure. The 2-DOF control system for the coarse stage and 6-DOF control system are coupled and combined, which is not allowed in the platform. In reference [25], a platform with three layers was designed. It can be used in sub-micrometer accuracy for nanotechnology applications; the stroke is 50 × 50 mm in the xy-motion. The air bearing was also used in the platform, which led to low stiffness. In reference [26], an array of cylindrical actuation coils and three position-sensing photodiode assemblies for the controller levitation were used in a magnetic suspension platform. With the novel structure, the platform’s stroke can be reached 30 mm horizontally and 20 mm vertically. In reference [27], the error budget methodology is used to improve the final system performance in the platform. By this method, the positioning error is less than 41, 36, and 48 nm on the *X*-, *Y*-, and *Z*-axes, respectively.

The traditional magnetic suspension platform has some shortcomings, such as short strokes, no gravity compensation mechanisms, no high start/stop acceleration, and low air gap magnetic flux density; however, the above characteristics are needed if the platform is used in mass transfer. In the paper, a novel magnetic suspension platform with an improved permanent magnetic array is proposed. Four passive magnetic bearings were used to compensate for the platform mass. The platform has the following merits: a simple structure, high force coefficient, and torque output, large force density, and a low driving force fluctuation. This paper is organized as follows. The following section introduces the structure and working principle of the proposed Lorentz force driving actuator and the platform. In Section 3, the suspension force model and driving force model are established by the subdomain method. In Section 4, the position measurement model and acceleration model of the actuator are established. In Section 5, to highlight the superiority of the proposed novel’s Halbach array structure, the magnetic flux density in the air gap and the driving force are investigated and compared by the finite element method. To show the merits of the proposed Lorentz force driving actuator, the driving force performances are compared with traditional ones.

## 2. Structure and Working Principle

The structure of the novel six-degree-of-freedom magnetic suspension platform is depicted in Figure 1; the novel platform in this paper is mainly composed of the stator and the Lorentz force actuator. The Lorentz force actuator includes driving coils, suspension coils, permanent magnets, displacement sensors, and the moving platen. The six-DOF motion is guaranteed by the Lorentz force actuator. Four driving coils were installed under the moving platen for *x*, *y* directions and the horizontal rotation. Four suspension coils were installed in the moving platen for moving in the *z*-direction and *z*-axis rotation. Four passive magnetic bearings were used as a gravity compensator. The stator’s magnetic array in this platform, different from the existing magnetic array in the stator, consists of a permanent magnetic and magnetizer. Figure 2 shows the working principle of the novel platform. The eight driving coils carry the direct current and are exposed to the magnetic field generated by the stator. Four suspension coils carry the direct current and are also exposed to the magnetic field produced by the stator. The red arrow is the direction of the driving force that the vertical driving coils can provide, and the blue arrow is the direction of the driving force that the horizontal driving coils can provide. Therefore, the motion state of the platform is changed by changing the direction of the direct current in the coils.

Figure 3 shows the structure of the Lorentz force actuator. The driving coils are fixed to the coil support frame, which is made of magnetic isolation material. The suspension coil support frames are made by a magnetizer, which can enhance the magnetic field generated by the current. The driving force and torque are controlled by changing the direct current in the driving coils. For instance, the driving force is outputted by driving coils 1 and 7 or 4 and 11 together. The torque is outputted by driving coils 1 and 7 or 4 and 11 together. The same working principle is conducted by the suspension coils. The mapping relationship between the platform motion and the driving/suspension coils of the 6-DOF of the platform is shown in Table 1.

## 3. Force Model

### 3.1. Suspension Force Model

To overcome the mass of the platform and obtain a stable suspension force, the passive magnetic bearings are used in platforms. The structure of the passive magnetic bearing is shown in Figure 4. In order to improve the magnetic flux density in the air gap, we used the novel permanent magnet array (PM array) along the *x*-direction, as shown in Figure 4c. To solve the magnetic field distribution, the PM arrays were divided into several regions according to the distribution law, as shown in Figure 4d.

In one divided area, the relationship between the magnetomotive force $\vec{F_{m}}$ and the magnetic flux ϕ is satisfied as follows:(1)$$\begin{array}{l} \phi =\vec{B}A_{m}\\ \vec{F_{m}}=\vec{H}l_{m} \end{array}$$
where *A_m_* is the magnetization direction area, *l_m_* is the length of the magnetization direction.

The relationship between $\vec{B}$ and the $\vec{H}$ is:(2)$$\begin{array}{l} \vec{B}=\mu_{0}\vec{H}\\ \vec{B}=\mu_{0}\mu_{r}\vec{H}+\mu_{0}\vec{M_{r}} \end{array}$$
where $\mu_{r}$ is the relative magnetic permeability, $\mu_{0}$ is the vacuum magnetic permeability, and $\vec{M_{r}}$ is the residual magnetization of the permanent magnet. 

As the linear demagnetization characteristic permanent magnet was used in the platform, so the $\vec{M_{r}}$ can be expressed as:(3)$$M_{r}=\frac{B_{r}}{\mu_{0}}$$

For this platform, the $\vec{M_{r}}$ is determined by the demagnetization direction and the PM array. So, the $\vec{M_{r}}$ in the cartesian coordinate system can be expressed as:(4)$$\begin{array}{l} \vec{M_{r1}}=\vec{M_{x1}}+\vec{M_{y1}}\\ \vec{M_{r2}}=\vec{M_{x1}}+\vec{M_{x2}}+\vec{M_{y1}}+\vec{M_{y2}} \end{array}$$
where the $\vec{M_{r1}}$ is region 1 $\vec{M_{r}}$, and the $\vec{M_{r2}}$ is region 2 $M_{r}$.

$\vec{M_{x1}}$, $\vec{M_{y1}}$, $\vec{M_{x2}}$, and $\vec{M_{y2}}$ are described by the Fourier series.
(5)$$\left\{ \begin{array}{l} \vec{M_{x1}}=\frac{a_{M01}}{2}+\sum_{n=1}^{\infty }a_{Mn1}\cos\left(k\omega x\right)\\ \vec{M_{y1}}=\sum_{n=1}^{\infty }b_{Mn1}\sin\left(k\omega x\right) \end{array}\right.$$
(6)$$\left\{ \begin{array}{l} \vec{M_{x2}}=\frac{a_{M02}}{2}+\sum_{n=1}^{\infty }a_{Mn2}\cos\left(k\omega x\right)\\ \vec{M_{y2}}=\sum_{n=1}^{\infty }b_{Mn2}\sin\left(k\omega x\right) \end{array}\right.$$
where:$$\begin{array}{l} a_{M01}=\frac{2M_{r1}\omega_{1}}{T},k_{1}=\frac{n_{1}\pi }{T}\\ a_{Mn1}=\frac{2M_{r1}}{n_{1}\pi }\left\{\sin\left[k_{1}(2\omega_{1}+3\omega_{2})\right]-\sin\left[k_{1}(2\omega_{1}+\omega_{2})\right]-\sin k_{1}\omega_{2}\right\}\\ b_{Mn1}=\frac{2M_{r1}}{n_{1}\pi }\left\{{\cos k}_{1}l-\cos\left[k_{1}(2\omega_{1}+3\omega_{2})\right]-\cos\left[k_{1}(2\omega_{1}+\omega_{2})\right]+\cos k_{1}\omega_{2}\right\}\\ a_{M02}=\frac{2M_{r2}\omega_{2}}{T},k_{2}=\frac{n_{2}\pi }{T}\\ a_{Mn2}=\frac{2M_{r2}}{n\pi }\left\{\sin\left[k_{2}(2\omega_{2}+3\omega_{1})\right]-\sin\left[k_{2}(2\omega_{2}+\omega_{1})\right]-\sin k_{2}\omega_{1}\right\}\\ b_{Mn2}=\frac{2M_{r2}}{n_{2}\pi }\left\{{\cos k}_{2}l-\cos\left[k_{2}(2\omega_{2}+3\omega_{1})\right]-\cos\left[k_{2}(2\omega_{2}+\omega_{1})\right]+\cos k_{2}\omega_{1}\right\} \end{array}$$

The magnetic vectors for region 1 and region 2 can be written as:(7)$$\begin{array}{l} A_{1}=\sum_{n=0}^{\infty }\left[\left(A_{n1}e^{ky}+B_{n1}^{-ky}\right)\cos\left(k_{1}x\right)+\left(C_{n1}e^{ky}+D_{n1}e^{-ky}\right)\sin\left(k_{1}x\right)\right]\\ A_{2}=\sum_{n=0}^{\infty }\left[\left(A_{n2}e^{ky}+B_{n2}^{-ky}\right)\cos\left(k_{2}x\right)+\left(C_{n2}e^{ky}+D_{n2}e^{-ky}\right)\sin\left(k_{2}x\right)\right] \end{array}$$

As the magnetic flux density is continuous anywhere, we can obtain the following formula:(8)$$\begin{array}{l} n\times \left(\vec{B_{1}}+\vec{B_{2}}\right)=0\\ n\times \left(\vec{H_{1}}+\vec{H_{2}}\right)=J\times S\\ \vec{B}=\nabla \times \vec{A}\\ \nabla \times \vec{H}=J \end{array}$$

With Equations (1)–(8), the flux density in every region can be calculated. We executed the finite element method before adopting the derived theoretical equation. The results of the two methods were obtained by the finite element method. The results show that the finite element method’s result is consistent with the mathematical models and the mathematical model can be used to describe the magnetic field of the novel PM array.

The repulsive force between the permanent magnets is used in the passive magnetic bearings, so the suspension force model for the platform can be written as:(9)$$F=\frac{1.5}{1+aL}{\left(\frac{B}{4865}\right)}^{2}A_{pm}$$
where *A_pm_* is the permanent magnetic area of the permanent magnet. Moreover, we can see from the suspension force model that the capacity of the platform can be improved by increasing the permanent magnetic area in the passive magnetic bearing.

### 3.2. Driving Force Model

The positioning accuracy and motion characteristics of the platform are determined by the driving force. The Lorentz coils are the sources of the driving forces, so the positions of the Lorentz coils should be first made. Then the actuator’s position can be determined by the positions of the Lorentz coils. The Lorentz coil positions are provided in Figure 5. The O-XYZ is a fixed coordinate system, and all the Lorentz coil positions can be described with O_i_-XYZ. The relationship between them is shown as follows:
(10)po=pi+poipoi=[xyz]
where po is the O-XYZ origin position, pi is the *i*-th Lorentz coil’s origin position, and the poi is the translation vector from pi to po.

When the actuator moves to a new position, the O-XYZ position is not changed, so we only need to solve the new position of the Lorentz coil. The new relationship between them is shown as follows:(11)po=Roiopi+poi

As shown in Figure 5, the new position of pi can be decomposed as a rotation around the *X*-, *Y*-, *Z*-axis and a translating vector of poi. The Roio can be expected as:(12)Roioi=ROz(α)ROy(β)ROx(γ)
where:(13)$$\begin{array}{l} R_{Oz}(\alpha )=\left[\begin{array}{ccc} \cos\alpha & -\sin\alpha & 0\\ \sin\alpha & \cos\alpha & 0\\ 0 & 0 & 1 \end{array}\right]\\ R_{Oy}(\beta )=\left[\begin{array}{ccc} \cos\beta & 0 & \sin\beta \\ 0 & 1 & 0\\ -\sin\beta & 0 & \cos\beta \end{array}\right]\\ R_{Ox}(\gamma )=\left[\begin{array}{ccc} 1 & 0 & 0\\ 0 & \cos\gamma & -\sin\gamma \\ 0 & \sin\gamma & \cos\gamma \end{array}\right] \end{array}$$

Then the relationship between the Lorentz coil’s new coordinate to the fixed coordinate O-XYZ can be expressed as:(14)BOiO=[cosαcosβcosαsinβsinγ−sinαcosγcosαsinβcosγ+sinαsinγxsinαcosβsinαsinβsinγ+cosαcosγsinαsinβcosγ−cosαsinγy−sinβcosβsinγcosβcosγz0001]
where (*x*,*y*,*z*) is the vector from O-XYZ to the new position of the Lorentz coil coordinate; α,β,γ are the rotations of the Lorentz coil coordinate revolving around the fixed coordinate O-XYZ.

When the auctor moves, the speed of the *i*-th Lorentz coil can be expressed as follows:(15)viO=voiO+BOiOvOiOiωiO=ωOiO×BOiOωOiOi
where vOi is the velocity of the *i*-th Lorentz coil relative to the fixed coordinate O-xyz, vOOi is the velocity of the *i*-th Lorentz coil. Moreover, so is ωOi, ωOOi. Then Equation (15) can be expressed as:(16)[viOωiO]=[BOiO00BOiO][vOiOiωOiOi]

Equation (16) is described by the Jacobian matrix:(17)JO(Θ)=[BOiO00BOiO]JOi(Θ)

The driving force model can be obtained based on Newton’s second law, as follows:(18)[FOMOi]=[BOiO0poiBOiO][FOiOiMOiOi]

It can be seen from Equation (18) that the Lorentz coil force is relative to the actuator. As shown in Figure 2, we take the *i*-th Lorentz coil as the analysis object to conduct the force current model. When the current is directed in a right-handed spiral with respect to the *X*-axis of the single coil, the Lorentz force $F_{i}$ can be expressed as:(19)$$F_{i}=\sum_{n=1,3}\int \int \int J_{n}\times \vec{B}dV_{n}+\sum_{n=2,4}\int \int \int J_{n}\times \vec{B}dV_{n}$$
where $\vec{B}$ is the magnetic flux density in the *i*-th Lorentz coil, $\vec{J_{n}}$ is the volume current density of the *i*-th Lorentz coil of the I array of the winding. The coils are symmetrical about the central plane; thus, (19) will be expressed as:(20)$$F_{i}=\sum_{n=1,3}\int \int \int J_{n}\times \vec{B}dV_{n}$$

The driving force of the all-Lorentz coils is expressed as:(21)$$F_{Oi}=\sum_{p=1}^{q}\sum_{m=1}^{z}{\sum_{n=1,3}\int \int \int J_{n}\times \vec{B}dV_{n}}_{i}$$
where *q* is the number of energized coils, *z* is the number of coil turns.

Based on the Lorentz force coil model shown in Figure 6, the Lorentz force torque is calculated in the fixed coordinate O-XYZ.

Taking the *i*-th and *j*-th Lorentz force coils as the analysis objects, the Lorentz force torque, $\vec{M_{ij}}$ around the origin of the fixed coordinate system, $O_{I}$, of the actuator is expressed as:(22)$$\begin{array}{l} \vec{M_{\mathrm{ijx}}}=\sum_{n=1,3}\int \int \int \vec{r_{13x}}\times J_{n}\times \vec{B}dV_{n}+\sum_{n=2,4}\int \int \int \vec{r_{24x}}\times J_{n}\times \vec{B}dV_{n}\\ \vec{M_{\mathrm{ijy}}}=\sum_{n=1,3}\int \int \int \vec{r_{13y}}\times J_{n}\times \vec{B}dV_{n}+\sum_{n=2,4}\int \int \int \vec{r_{24y}}\times J_{n}\times \vec{B}dV_{n} \end{array}$$
where $\vec{r_{13x}}$ is the position vector from the fixed coordinate O_i_-XYZ of the auctor to a point of the winding of the coils 1_x_ and 3_x_, $\vec{r_{24x}}$ is the position vector from the fixed coordinate O_i_-XYZ of the auctor to a point of the winding of the coils 2_x_ and 4_x_, $\vec{r_{13y}}$ is the position vector from the fixed coordinate O_i_-XYZ of the auctor to a point of the winding of the coils 1_y_ and 3_y,_
$\vec{r_{24y}}$ is the position vector from the fixed coordinate O_i_-XYZ of the auctor to a point of the winding of the coils 2_y_ and 4_y_.

$\vec{r_{13x}}$, $\vec{r_{24x}}$, $\vec{r_{13y}}$, $\vec{r_{24y}}$ can be denoted as:(23)$$\begin{array}{l} \vec{r_{13x}}=\vec{O_{3x}O_{i}}+\vec{O_{i}O_{1x}}+\vec{O_{1x}R_{1x}}+\vec{O_{3x}R_{3x}}\\ \vec{r_{24x}}=\vec{O_{2x}O_{i}}+\vec{O_{i}O_{2x}}+\vec{O_{2x}R_{2x}}+\vec{O_{4x}R_{4x}}\\ \vec{r_{13y}}=\vec{O_{3y}O_{i}}+\vec{O_{i}O_{1y}}+\vec{O_{1y}R_{1y}}+\vec{O_{3y}R_{3y}}\\ \vec{r_{24y}}=\vec{O_{2y}O_{i}}+\vec{O_{i}O_{2y}}+\vec{O_{2y}R_{2y}}+\vec{O_{4y}R_{4y}} \end{array}$$

Submitting (23) into (22), the torque, $\vec{M_{ij}}\;$ is simplified to:(24)$$\begin{array}{l} \vec{M_{ijx}}=\vec{N_{ijx}}J_{n}\\ \vec{M_{ijy}}=\vec{N_{ijy}}J_{n} \end{array}$$

According to Newton’s law, the driving torque acting on the actuator can be expressed as:(25)$$M=\vec{M_{ijx}}+\vec{M_{ijy}}$$

## 4. Measurement Model of Actuator

The novel 6-DOF magnetic suspension platform has the following advantages: large strokes, ultra-precision, high start/stop acceleration, and low power consumption. The actuator’s position information is important for determining the stroke, acceleration, and velocity of the platform, which can be obtained by the displacement sensors. In the active control system, obtaining the position of the actuator is not only the most direct method to measure the speed and acceleration of the platform, but it also provides the feedback signal for the control system, which is vital for the stability of the platform.

The three-dimensional views of the proposed novel 6-DOF magnetic suspension platform are provided in Figure 7; twelve displacement sensors are fixed at different positions of the actuator. A control circuit board is fixed in the actuator to process the real-time information of the actuator and to control the current of the Lorentz coils. Four sensors are ZH52A0801; they are differential eddy current displacement sensors with a range of ∓4 mm and a resolution of 5 nm. The other displacement sensors are BX-LV400N; they are laser displacement sensors with a range of ∓600 mm and a resolution of 8 μm. As the resolution of BX-LV400N cannot meet the position requirements, a control method is needed to achieve the nano-position accuracy, which will be explained in other papers.

### 4.1. Position Measurement Model for the Actuator

The actuator’s position and attitude are determined by the Lorentz coils and the fixed coordinate O-XYZ of the stator. The positions and attitude models of the actuator are established. 

The original relative positions of the Lorentz coils $O_{i}\mathrm{XYZ}$(I = 1, 2…, 12) to the fixed coordinates O-XYZ are shown in Figure 5. As the Lorentz coils are placed symmetrically, it is not necessary to study the motion law of all coil coordinates. The $O_{\mathrm{lci}}\mathrm{XYZ}$ (I = 1…, 6) will be studied and analyzed when the actuator moves. The actuator in the original place, the $O_{\mathrm{lci}}\mathrm{XYZ}$, can be expressed as:(26)$$\begin{array}{l} O_{lc1}=\left[\begin{array}{c} a+l_{1}\\ b\\ 0 \end{array}\right]\;O_{lc2}=\left[\begin{array}{c} a\\ b+l_{2}\\ 0 \end{array}\right]O_{lc3}=\left[\begin{array}{c} a\\ b\\ 0 \end{array}\right]\\ O_{lc4}=\left[\begin{array}{c} -a-l_{1}\\ b\\ 0 \end{array}\right]\;O_{lc5}=\left[\begin{array}{c} -a\\ b+l_{2}\\ 0 \end{array}\right]\;O_{lc6}=\left[\begin{array}{c} -a\\ b\\ 0 \end{array}\right] \end{array}$$
where a is the distance from $O_{lc1}$ to the *Y*-axis of the O-XYZ coordinate, *b* is the distance from $O_{lc1}$ to the *X*-axis of the O-XYZ coordinate, $l_{1}$ is the distance from $O_{lc1}$ to $O_{lc2}$, $l_{2}$ is the distance from $O_{lc1}$ to $O_{lc3}$.

When the actuator moves to a new place, the $O_{lci}XYZ$ will be expressed as:(27)$$\begin{array}{l} O_{lc1}=\left[\begin{array}{c} a+l_{1}+x\\ b+y\\ z_{1} \end{array}\right]\;O_{lc2}=\left[\begin{array}{c} a+x\\ b+l_{2}+y\\ z_{2} \end{array}\right]O_{lc3}=\left[\begin{array}{c} a+x\\ b+y\\ z_{3} \end{array}\right]\\ O_{lc4}=\left[\begin{array}{c} -a-l_{1}+x\\ b+y\\ z_{4} \end{array}\right]\;O_{lc5}=\left[\begin{array}{c} -a+x\\ b+l_{2}+y\\ z_{5} \end{array}\right]\;O_{lc6}=\left[\begin{array}{c} -a+x\\ b+y\\ z_{6} \end{array}\right] \end{array}$$

The rotation angle of the actuator around the fixed coordinate O-XYZ can be expressed as:(28)$$\begin{array}{l} \alpha =\arctan\frac{b+y}{a+l_{1}+x}\\ \beta =\arctan\frac{z_{1}}{\sqrt{{\left(b+y\right)}^{2}+{\left(a+l_{1}+x\right)}^{2}}}\\ \gamma =\arctan\frac{z_{4}}{\sqrt{{\left(-a-l_{1}+x\right)}^{2}+{\left(b+y\right)}^{2}}} \end{array}$$
where *x*, *y*, and *z* are the distances of the $O_{\mathrm{lc}1}\mathrm{XYZ}$ move relative to the fixed coordinates O-XYZ.

Submitting Equations (27) and (28) to Equation (14), the actuator’s new place can be expressed as:(29)$$\left[\begin{array}{cccc} a_{11} & a_{12} & a_{13} & x\\ a_{21} & a_{22} & a_{23} & y\\ a_{31} & a_{32} & a_{33} & z\\ 0 & 0 & 0 & 1 \end{array}\right]$$
where:$$\begin{array}{l} a_{11}=\frac{\left(a+l_{1}+x\right)\sqrt{{\left(b+y\right)}^{2}+{\left(a+l_{1}+x\right)}^{2}}}{\sqrt{{\left(a+l_{1}+x\right)}^{2}+{\left(b+y\right)}^{2}}\sqrt{z_{1}^{2}+{\left(b+y\right)}^{2}+{\left(a+l_{1}+x\right)}^{2}}}\\ a_{12}=\frac{z_{1}z_{4}\left(a+l_{1}+x\right)}{\sqrt{{\left(a+l_{1}+x\right)}^{2}+{\left(b+y\right)}^{2}}\sqrt{z_{1}^{2}+{\left(b+y\right)}^{2}+{\left(a+l_{1}+x\right)}^{2}}\sqrt{z_{4}^{2}+{\left(-a-l_{1}+x\right)}^{2}+{\left(b+y\right)}^{2}}}-\frac{(b+y)\sqrt{{\left(-a-l_{1}+x\right)}^{2}+{\left(b+y\right)}^{2}}}{\sqrt{{\left(a+l_{1}+x\right)}^{2}+{\left(b+y\right)}^{2}}\sqrt{z_{4}^{2}+{\left(-a-l_{1}+x\right)}^{2}+{\left(b+y\right)}^{2}}}\\ a_{13}=\frac{z_{1}\left(a+l_{1}+x\right)\sqrt{{\left(-a-l_{1}+x\right)}^{2}+{\left(b+y\right)}^{2}}}{\sqrt{{\left(a+l_{1}+x\right)}^{2}+{\left(b+y\right)}^{2}}\sqrt{z_{1}^{2}+{\left(b+y\right)}^{2}+{\left(a+l_{1}+x\right)}^{2}}\sqrt{z_{4}^{2}+{\left(-a-l_{1}+x\right)}^{2}+{\left(b+y\right)}^{2}}}-\frac{(b+y)\sqrt{{\left(-a-l_{1}+x\right)}^{2}+{\left(b+y\right)}^{2}}}{\sqrt{{\left(a+l_{1}+x\right)}^{2}+{\left(b+y\right)}^{2}}\sqrt{z_{4}^{2}+{\left(-a-l_{1}+x\right)}^{2}+{\left(b+y\right)}^{2}}}\\ a_{21}=\frac{(b+y)\sqrt{{\left(b+y\right)}^{2}+{\left(a+l_{1}+x\right)}^{2}}}{\sqrt{{\left(a+l_{1}+x\right)}^{2}+{\left(b+y\right)}^{2}}\sqrt{z_{1}^{2}+{\left(b+y\right)}^{2}+{\left(a+l_{1}+x\right)}^{2}}}\\ a_{22}=\frac{z_{1}z_{4}(b+y)}{\sqrt{{\left(a+l_{1}+x\right)}^{2}+{\left(b+y\right)}^{2}}\sqrt{z_{1}^{2}+{\left(b+y\right)}^{2}+{\left(a+l_{1}+x\right)}^{2}}\sqrt{z_{4}^{2}+{\left(-a-l_{1}+x\right)}^{2}+{\left(b+y\right)}^{2}}}-\frac{\left(a+l_{1}+x\right)\sqrt{{\left(-a-l_{1}+x\right)}^{2}+{\left(b+y\right)}^{2}}}{\sqrt{{\left(a+l_{1}+x\right)}^{2}+{\left(b+y\right)}^{2}}\sqrt{z_{4}^{2}+{\left(-a-l_{1}+x\right)}^{2}+{\left(b+y\right)}^{2}}}\\ a_{23}=\frac{z_{1}(b+y)\sqrt{{\left(-a-l_{1}+x\right)}^{2}+{\left(b+y\right)}^{2}}}{\sqrt{{\left(a+l_{1}+x\right)}^{2}+{\left(b+y\right)}^{2}}\sqrt{z_{1}^{2}+{\left(b+y\right)}^{2}+{\left(a+l_{1}+x\right)}^{2}}\sqrt{z_{4}^{2}+{\left(-a-l_{1}+x\right)}^{2}+{\left(b+y\right)}^{2}}}-\frac{z_{4}\left(a+l_{1}+x\right)}{\sqrt{{\left(a+l_{1}+x\right)}^{2}+{\left(b+y\right)}^{2}}\sqrt{z_{4}^{2}+{\left(-a-l_{1}+x\right)}^{2}+{\left(b+y\right)}^{2}}}\\ a_{31}=-\frac{z_{1}}{\sqrt{z_{1}^{2}+{\left(b+y\right)}^{2}+{\left(a+l_{1}+x\right)}^{2}}}\\ a_{32}=\frac{z_{4}\sqrt{{\left(b+y\right)}^{2}+{\left(a+l_{1}+x\right)}^{2}}}{\sqrt{z_{1}^{2}+{\left(b+y\right)}^{2}+{\left(a+l_{1}+x\right)}^{2}}\sqrt{z_{4}^{2}+{\left(-a-l_{1}+x\right)}^{2}+{\left(b+y\right)}^{2}}}\\ a_{33}=\frac{\sqrt{{\left(b+y\right)}^{2}+{\left(a+l_{1}+x\right)}^{2}}\sqrt{{\left(-a-l_{1}+x\right)}^{2}+{\left(b+y\right)}^{2}}}{\sqrt{z_{1}^{2}+{\left(b+y\right)}^{2}+{\left(a+l_{1}+x\right)}^{2}}\sqrt{z_{4}^{2}+{\left(-a-l_{1}+x\right)}^{2}+{\left(b+y\right)}^{2}}} \end{array}$$

To improve the accuracy of the actuator’s position model, the average values of x, y, and z are used.

### 4.2. Acceleration Model of Actuator

The novel magnetic suspension in this paper, different from the existing magnetic suspension platform, has large strokes, large start/stop acceleration, and ultra-precision. Thus, it was necessary to measure the acceleration and the speed of the platform in real-time. The real-time displacement was obtained by the displacement sensors and the actuator’s velocity and acceleration were solved by a control program. For simplification, the actuator’s velocity is expressed by the coordinate $O_{a}\mathrm{XYZ}$. The $O_{a}\mathrm{XYZ}$ velocity is expressed as:(30)$$\left[\begin{array}{c} v_{ao}\\ \omega_{ao} \end{array}\right]=\left[\begin{array}{c} v_{aox}\\ v_{aoy}\\ v_{aoz}\\ \omega_{aox}\\ \omega_{aoy}\\ \omega_{aoz} \end{array}\right]=I\left[\begin{array}{c} x{\left(t\right)}_{aox}\\ y{\left(t\right)}_{aoy}\\ z{\left(t\right)}_{aoz}\\ \omega {\left(t\right)}_{aox}\\ \omega {\left(t\right)}_{aoy}\\ \omega {\left(t\right)}_{aoz} \end{array}\right]$$
where *I* is the identity matrix, *x*, *y*, *z,* ω are functions of time t.

With the displacement sensor measurement values, the actuator’s velocity can be expressed as:(31)$$\left[\begin{array}{c} v_{ao}\\ \omega_{ao} \end{array}\right]=I\left[\begin{array}{c} x{\left(t\right)}_{aox}\\ y{\left(t\right)}_{aoy}\\ z{\left(t\right)}_{aoz}\\ \omega {\left(t\right)}_{aox}\\ \omega {\left(t\right)}_{aoy}\\ \omega {\left(t\right)}_{aoz} \end{array}\right]=\left[\begin{array}{c} \frac{x_{lc1}+x_{lc2}+x_{lc4}+x_{lc5}+x_{lc7}+x_{lc8}+x_{lc10}+x_{lc11}}{8t}\\ \frac{y_{lc1}+y_{lc2}+y_{lc4}+y_{lc5}+y_{lc7}+y_{lc8}+y_{lc10}+y_{lc11}}{8t}\\ \frac{z_{lc3}+z_{lc6}+z_{lc9}+z_{lc11}}{4t}\\ \frac{1}{2t}\arctan\frac{\left|z_{lc1}\right|-\left|z_{lc4}\right|}{\left|x_{lc1}-z_{lc4}\right|}+\frac{1}{2t}\arctan\frac{\left|z_{lc11}\right|-\left|y_{lc7}\right|}{\left|x_{lc11}-x_{lc7}\right|}\\ \frac{1}{2t}\arctan\frac{\left|z_{lc5}\right|-\left|z_{lc2}\right|}{\left|y_{lc5}-y_{lc2}\right|}+\frac{1}{2t}\arctan\frac{\left|z_{lc8}\right|-\left|z_{lc10}\right|}{\left|y_{lc8}-y_{lc10}\right|}\\ \frac{1}{2t}\arctan\frac{y_{lc5}-y_{lc2}}{\left|x_{lc5}\right|-\left|x_{lc2}\right|}+\frac{1}{2t}\arctan\frac{y_{lc4}-y_{lc1}}{\left|x_{lc4}\right|-\left|x_{lc1}\right|} \end{array}\right]$$

With Equation (31), the acceleration of the platform can be expressed as:(32)$$a=\left[\begin{array}{c} a_{v}\\ a_{\omega } \end{array}\right]={\left[\begin{array}{c} v_{ao}\\ \omega_{ao} \end{array}\right]}^{\prime }=I{\left[\begin{array}{c} x{\left(t\right)}_{aox}\\ y{\left(t\right)}_{aoy}\\ z{\left(t\right)}_{aoz}\\ \omega {\left(t\right)}_{aox}\\ \omega {\left(t\right)}_{aoy}\\ \omega {\left(t\right)}_{aoz} \end{array}\right]}^{\prime }={\left[\begin{array}{c} \frac{x_{lc1}+x_{lc2}+x_{lc4}+x_{lc5}+x_{lc7}+x_{lc8}+x_{lc10}+x_{lc11}}{8t}\\ \frac{y_{lc1}+y_{lc2}+y_{lc4}+y_{lc5}+y_{lc7}+y_{lc8}+y_{lc10}+y_{lc11}}{8t}\\ \frac{z_{lc3}+z_{lc6}+z_{lc9}+z_{lc11}}{4t}\\ \frac{1}{2t}\arctan\frac{\left|z_{lc1}\right|-\left|z_{lc4}\right|}{\left|x_{lc1}-z_{lc4}\right|}+\frac{1}{2t}\arctan\frac{\left|z_{lc11}\right|-\left|y_{lc7}\right|}{\left|x_{lc11}-x_{lc7}\right|}\\ \frac{1}{2t}\arctan\frac{\left|z_{lc5}\right|-\left|z_{lc2}\right|}{\left|y_{lc5}-y_{lc2}\right|}+\frac{1}{2t}\arctan\frac{\left|z_{lc8}\right|-\left|z_{lc10}\right|}{\left|y_{lc8}-y_{lc10}\right|}\\ \frac{1}{2t}\arctan\frac{y_{lc5}-y_{lc2}}{\left|x_{lc5}\right|-\left|x_{lc2}\right|}+\frac{1}{2t}\arctan\frac{y_{lc4}-y_{lc1}}{\left|x_{lc4}\right|-\left|x_{lc1}\right|} \end{array}\right]}^{\prime \prime }$$

For the convenience of controlling, the linear acceleration and the angular acceleration of the platform can be expressed as (*z*-axis as an example):(33)aoz=(zlc3t1+zlc6t1+zlc9t1+zlc11t1)−(zlc3t2+zlc6t2+zlc9t2+zlc11t2)4Δtωoz={[12tarctanylc5t1−ylc2t1|xlc5t1|−|xlc2t1|+12tarctanylc4t1−ylc1t1|xlc4t1|−|xlc1t1|]−[12tarctanylc5t2−ylc2t2|xlc5t2|−|xlc2t2|+12tarctanylc4t2−ylc1t2|xlc4t2|−|xlc1t2|]}Δt

With the same method, the $a_{oy}$, $a_{ox}$, $\omega_{ax}$, $\omega_{ay}$ can be obtained. The platform’s acceleration can be run by the addition rule for vectors.

## 5. Suspension Force and Driving Force Analysis

The maxwell is used to analyze the magnetic circuit in 2D simplification. To achieve ultra-precision, large strokes, and high start/stop acceleration, the air gap’s magnetic flux density should have a large and uniform magnetic flux density (B). For the conventional magnetic suspension platform, the Halbach array and normal array were used. The traditional Halbach permanent magnet has a high B, but the magnetic density fluctuation is also large. The two methods above could not meet the novel magnetic suspension platform requirements. Thus, a novel permanent magnetic array was used. In the conventional magnetic array, the yoke is only used on one side to reduce the magnetic flux resistance and leakage. The main innovation of this paper was to use the yoke on three sides of the permanent magnetic. Small magnets were used to adjust the air gap magnetic density (different from the traditional Halbach array). This is the main difference between the two conventional schemes. The magnetic circuit of the permanent magnet is shown in Figure 8.

### 5.1. Analysis of the Magnetic Flux Density

The magnetic flux density is a key factor in the novel platform. The finite element method is used to analyze and compare the *B* of the Halbach array scheme and the novel array. As shown in Figure 9. The magnetization directions of the PMs are the same in the two schemes. However, the novel array has two sizes of PMs. The B is more uniform than the schemes that do not use the small PMs. From Equation (22), the air gap magnetic field characteristics determined the driving force and the driving torque of the Lorentz actuator. To achieve the goal of a large stroke, the magnetic flux density should have the same direction, which will decrease the B. The air gap magnetic density distribution is non-uniform in the two schemes, but the novel array has a better sinusoidal distribution. For the traditional Halbach array, the B is in the opposite direction in regions I and V, which is bad for the driving force. However, for the novel array, the opposite direction region is small, and it can be eliminated by setting the length of the Lorentz coils.

The air gap flux densities of the two schemes are depicted in Figure 10. It can be seen that the traditional array has a bigger B than the novel array but it has a large magnetic flux density fluctuation, which cannot be allowed in the large stroke magnetic suspension platform. For the novel array, the magnetic flux density is more uniform when the *Z* becomes large. Although the B decreases with the increase of the *Z*-axis distance, the driving force and output torque can be kept constant by increasing the current, which is easy to achieve.

The NdFeB35 is used in the novel platform. For a large and uniform flux density distribution in the air gap, the size of the PM is optimized by the finite element method. As shown in Figure 11, the air gap flux density will increase with the increase of the magnetization length. However, the air gap magnetic density will not change when the magnetization length of the permanent magnet exceeds a certain value. At the optimum magnetization length, the magnetic density at the center of the PM will decrease as the width of the PM increases. Moreover, the above problem can be solved by the novel Halbach PM array.

### 5.2. Analysis of the Force

The driving force is a key factor for the novel platform to achieve high acceleration and run smoothly. Moreover, the relation between the driving force and motion state is the key to building the control model. Under the 2A current, with the coil turning 500, the driving forces of the *X*-axis and *Y*-axis are analyzed by the finite element method. At the same current, the *X*-axis driving force and *Y*-axis driving force are almost the same because the Lorenz coils have the same sizes and the magnetic field is uniform. Figure 12 shows that the driving force in the *X*-axis produced by the Lorentz coils is not changed when the actuator moves. The average driving force of the novel Halbach array is about 63 N, which is 18.8% larger than 53 N of the traditional one. That is because the novel Halbach array magnetic flux density is more uniform than the traditional one, as shown in Figure 10. 

To analyze the stability of the novel platform, it was necessary to study the ripple ratio (*η*) of the driving force. So the *η* was introduced to evaluate the fluctuation of the driving force. It can be expressed as follows:(34)$$\eta =100\%\times \frac{F-F_{\mathrm{agv}}}{F_{agv}}$$

Figure 13 shows that the ripple ratio of the driving force is less than 4%, but the novel platform has a better *η*. The maximum *η* is 1.5%, which is 3.4% for the traditional Halbach array. The average *η* for the novel Halbach array is 0.53%, while the traditional Halbach array is 1.46%, so it is easier for the novel platform to realize stability.

The finite element method was also used to analyze the torque. As shown in Table 2. The output torque of the traditional Halbach array is 6.5 N*cm, while the novel Halbach array is 7.4 N*cm, which is better for the platform to rotate to a fixed angle in a short time. The large output torque is also suited for mass transfer.

## 6. Conclusions

A novel 6-DOF Lorentz force actuator with an improved Halbach array is proposed for the magnetic suspension platform. The translation stroke is 300 × 300 × 5 mm, the overall dimension is 600 × 600 × 45 mm, and the deflection is 5°. The suspension model, driving force model, the position measurement model, and the acceleration model were established by the subdomain method. Moreover, the structure of the platform is explained. The magnetic flux density, the driving force, and the output torque were analyzed by the finite element method. Compared with the traditional Halbach array, the maximum B in the air gap was 0.89T for the novel Halbach array and 0.83T in the traditional Halbach array. The average driving force in the *X*-axis was 63N; it was 53N in the traditional Halbach. The average force in the *Y*-axis was 61.3N; it was 51.8N in the traditional Halbach. The force improved significantly. The magnetic flux density’s uniform improved significantly. The maximum ripple ratio of the driving force was 1.5% in the novel platform; it was 3.4% for the traditional Halbach array. Compared with the traditional 6-DOF magnetic suspension platform, the platform in this paper is large, and its structure is novel. Passive magnetic bearings were first used in the platform, reducing the power consumption and coil heat.

## Figures and Tables

**Figure 1 sensors-22-04067-f001:**
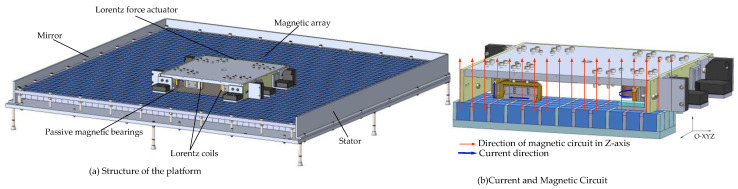
The structure of the 6-DOF magnetic suspension platform.

**Figure 2 sensors-22-04067-f002:**
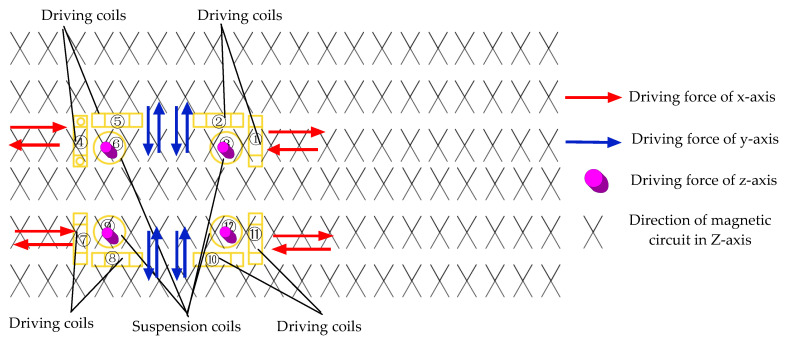
Working principle of the 6-DOF magnetic suspension platform.

**Figure 3 sensors-22-04067-f003:**
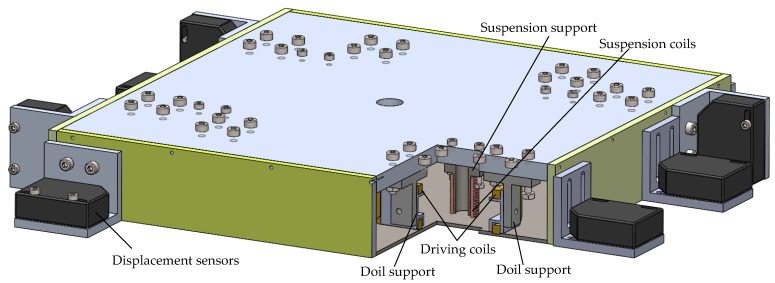
Structure of the 6-DOF magnetic suspension platform’s actuator.

**Figure 4 sensors-22-04067-f004:**
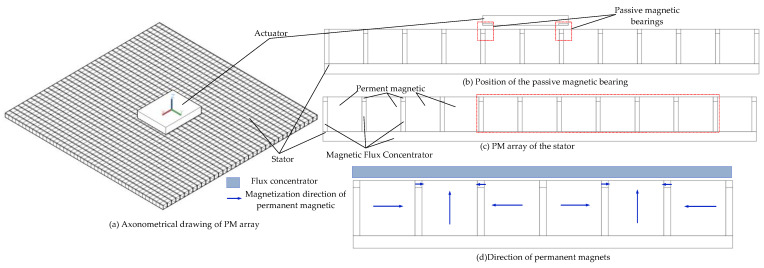
Structure of the PM array.

**Figure 5 sensors-22-04067-f005:**
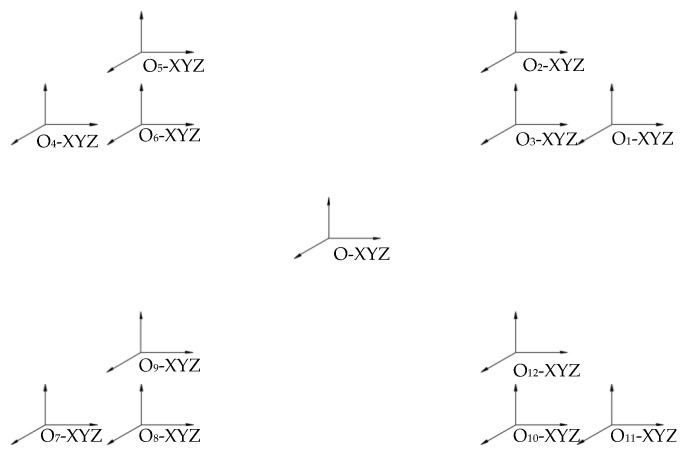
Coordinate system of a novel 6-DOF suspension platform.

**Figure 6 sensors-22-04067-f006:**
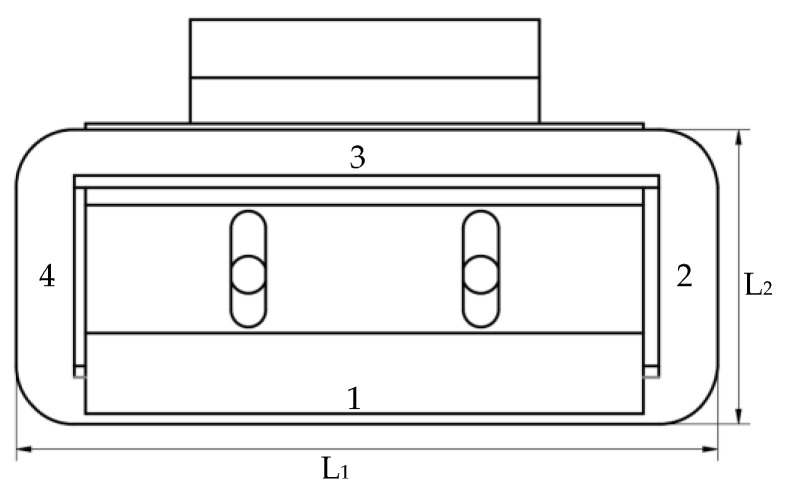
Model of the Lorentz coils.

**Figure 7 sensors-22-04067-f007:**
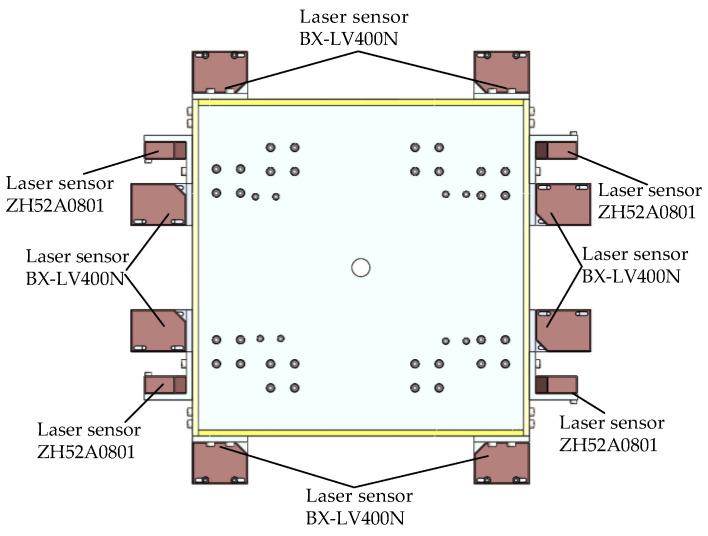
Sensor and controller on the 6-DOF platform actuator.

**Figure 8 sensors-22-04067-f008:**
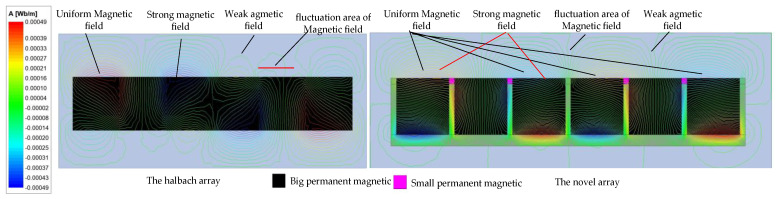
Magnetic circuit of the two schemes.

**Figure 9 sensors-22-04067-f009:**
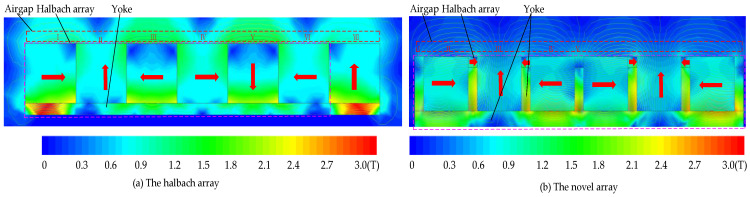
The magnetic density cloud map of the two schemes.

**Figure 10 sensors-22-04067-f010:**
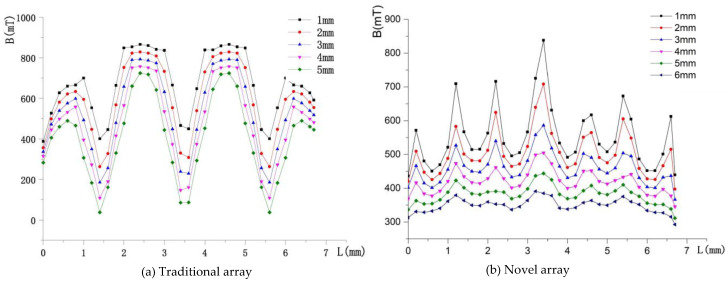
Air gap flux density distribution of the two schemes.

**Figure 11 sensors-22-04067-f011:**
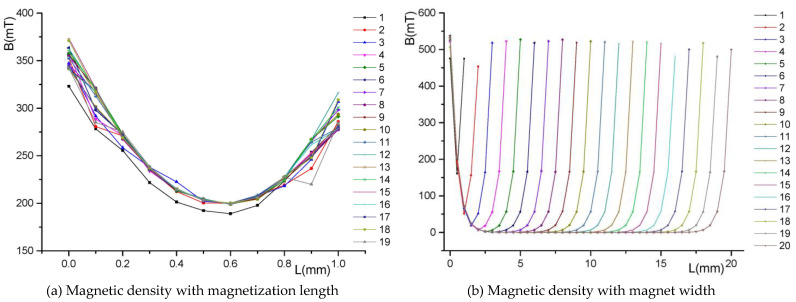
Relationship between the magnetic density and the size of the permanent magnet.

**Figure 12 sensors-22-04067-f012:**
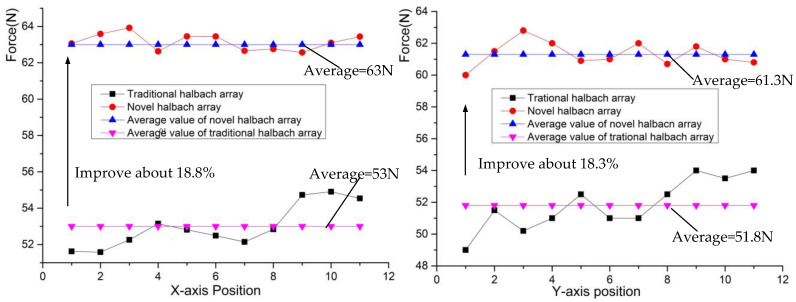
Relationship between the driving force and the actuator’s position.

**Figure 13 sensors-22-04067-f013:**
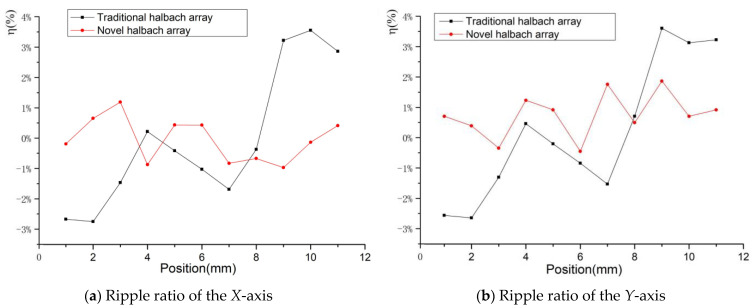
Ripple ratio of the driving force.

**Table 1 sensors-22-04067-t001:** The relationship between the motion of the platform and the driving/suspension coil.

Movement of the Platform	Driving Force/Torque
Translation in *X*-axis direction of O-XYZ	Driving coils 1 and 7 or 4 and 11
Translation in *Y*-axis direction of O-XYZ	Driving coils 5 and 10 or 2 and 8
Translation in *Z*-axis direction of O-XYZ	Suspension coils 3, 6, 9, 12
Rotate around *X*-axis of O-XYZ	Suspension coils 3, 6 and 9, 12
Rotate around *Y*-axis of O-XYZ	Suspension coils 3 12 and 6, 9
Rotate around *Z*-axis of O-XYZ	Driving coils 1 and 7 or 4 and 11 or 5 and 10 or 2 and 8

**Table 2 sensors-22-04067-t002:** The output torques of the two types of platforms.

	The Halbach Array	The Novel Array
Value	6.5 N*cm	7.4 N*cm

## Data Availability

Not applicable.

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
