# Peer review of "Design and Analysis of a 6-DOF Magnetic Suspension Platform with an Improved Permanent Magnetic Array"

_sensors, 2022, doi:10.3390/s22114067_

Round 1

Reviewer 1 Report

The font and size should be uniform throughout.

Please use the full name and center of the axis even if the axis title is mentioned in the text.

figure2: The text is hard to distinguish because of the background.

figure3: The text is too small.
figure4: I can't tell the difference between a magnet and a flux concentrator. It would be good to make the distinction easier by adding a soft color, etc.
figure8: Lines are not well separated. Also, there is no clear distinction between a weak magnetic field and a strong magnetic field. Only the directions are different. As mentioned in the text, whether the magnetic flux is more uniform is not distinguishable. Are the magnetic flux scales in the two figures the same? You need to reorganize the figure clearly as a whole.

figure9: You should use the same color scale for precise comparison.
And it seems that the terminology needs to be unified with figure8.

figure12: plotting the mean as a line seems to make the distinction clearer.

Some equations are too large and complex. (ex. Eq 29)
It is necessary for calculation, but it is questionable whether it is an essential part of the paper. If possible, it is better to be a little more concise.

Author Response

(Note from the assistant editor: Dear Reviewer, you are marked as Reviewer 1 in the system. Please kindly check the authors' response letter.)

Reviewer 2 Report

The paper can be accepted after the following corrections:

  1. line 203: please correct the equation
  2. Figure 1, 2 and 3 high quality, but they are not clear from technical point of view. Please re-draw to clearly present and explain magnet orientation as well as the principles of operation of both platform and the actuator.
  3. Equations (12) and (13) are trivial. One of them should be removed.
  4. The photography of "Controller of the 6-DOF platform" is not suitable for the scientific publication. Please remove.
  5. More details about the FEM analyses (section 5) should be provided. What were the sotware used? Was it 2D simplification of 3D system?
  6. Please develop the conclusions, to present summary and advantages of proposed system in more quantitative way.

Author Response

(Note from the assistant editor: Dear Reviewer, you are marked as Reviewer 2 in the system. Please kindly check the authors' response letter.)

Reviewer 3 Report

The authors present a design and analysis of a 6-DOF magnetic suspension platform with an improved permanent magnetic array. An accurate model of the novel structure is established to calculate the magnetic density distribution for obtaining the parameters and performance of the magnetic suspension platform. The results obtained by analytical models are verified by using finite element method. The obtained results are original and could be interesting for the readers of the Sensors journal, however, the paper could not be published in the present form.

Comments:

  1. The manuscript has a lot of spelling mistakes. The most common are dots or comma not in the right place or after the dot the sentence starts with a lowercase letter, for example lines 124, 162, 364, 376, 386. Many not clear sentences such as: p13, line 347 “The traditional Halbach array has a large but shows a large fluctuation B.“
  2. The manuscript is difficult to read because of the large number of formulas that should be moved to the appendix or supplement.
  3. The manuscript lacks simulation of platform movement, there are presented only the static data.
  4. The authors also write nothing about inertia during the motion of a given platform and how it is compensated.
  5. It is not clear, what is shown in Figure 6. Please explain in more details.
  6. There is a lack of comparison of the obtained results with the data from the literature.
  7. In the introduction, the authors highlight the low movement speed of such platforms as one of the problems. However, it is not clearly explained if these problems are solved in this work. The same problem is with the accuracy of platform positioning. Although the authors in their simulation model use a fairly accurate positioning gauge, but it is not clear how it would be in reality.
  8. Figure 7 shows a real schematic realization of the controller on the PCB. Does it mean that such a platform exists in reality? If so, why the results of experiments are not given?
  9. Error in Figure 8: The position of the magnetic flux concentrator is not correctly shown. Also, it is not shown (in this figure) where the additional small magnets are placed.
  10. Figure 14 makes no sense at all. The authors can give some numbers in the text showing improvement of the proposed platform.

Author Response

(Note from the assistant editor: Dear Reviewer, you are marked as Reviewer 3 in the system. Please kindly check the authors' response letter.)

Round 2

Reviewer 3 Report

The authors corrected the manuscript according to the comments. Therefore, I suggest to accept in present form.